# Ternary Weight Decomposition and Binary Activation Encoding for Fast and Compact Neural Network

**Mitsuru Ambai & Takuya Matsumoto**
Denso IT Laboratory, Inc.
{manbai,tmatsumoto}@d-itlab.co.jp

**Takayoshi Yamashita & Hironobu Fujiyoshi**
Chubu University
{yamashita,hf}@cs.chubu.ac.jp

## Abstract

This paper aims to reduce test-time computational load of a deep neural network. Unlike previous methods which factorize a weight matrix into multiple real-valued matrices, our method factorizes both weights and activations into integer and non-integer components. In our method, the real-valued weight matrix is approximated by a multiplication of a ternary matrix and a real-valued co-efficient matrix. Since the ternary matrix consists of three integer values, $\{-1, 0, +1\}$, it only consumes 2 bits per element. At test-time, an activation vector that passed from a previous layer is also transformed into a weighted sum of binary vectors, $\{-1, +1\}$, which enables fast feed-forward propagation based on simple logical operations: AND, XOR, and bit count. This makes it easier to deploy a deep network on low-power CPUs or to design specialized hardware.

In our experiments, we tested our method on three different networks: a CNN for handwritten digits, VGG-16 model for ImageNet classification, and VGG-Face for large-scale face recognition. In particular, when we applied our method to three fully connected layers in the VGG-16, $15\times$ acceleration and memory compression up to $5.2\%$ were achieved with only a $1.43\%$ increase in the top-5 error. Our experiments also revealed that compressing convolutional layers can accelerate inference of the entire network in exchange of slight increase in error.

## 1 Introduction

It is widely believed that deeper networks tend to achieve better performance than shallow ones in various computer vision tasks. As a trade-off of such impressive improvements, deeper networks impose heavy computational load both in terms of processing time and memory consumption due to an enormous amount of network parameters. For example, VGG-16 model (Simonyan & Zisserman, 2015) requires about 528 MBytes to store the network weights where fully connected layers account for $89\%$ of them. A large number of multiplications and additions must also be processed at each layer which prevent real-time processing, consume vast amounts of electricity, and require a large number of logic gates when implementing a deep network on a FPGA or ASIC.

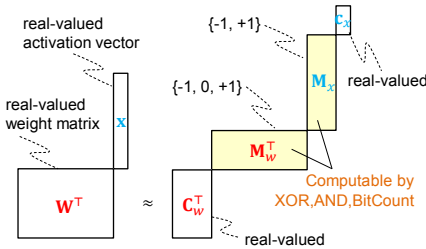

Figure 1: Our network compression model

This article addresses the above issues. Specifically, we aimed to reduce the test-time computational load of a pre-trained network. Since our approach does not depend on a network configuration (e.g. a choice of an activation function, layer structures, and a number of neurons) and acts as a post-processing of network training, pre-trained networks shared in a download site of MatConvNet (Vedaldi & Lenc, 2015) and Model Zoo (BVLC) can be compressed and accelerated. Our method is outlined in Figure 1. The main idea is to factorize both weights and activations into integer and non-integer components. Our method is composed of two building blocks, as shown below.

**Ternary weight decomposition for memory compression**: We introduce a factored representation where the real-valued weight matrix is approximated by a multiplication of a ternary basis matrix and a real-valued co-efficient matrix. While the ternary basis matrix is sufficiently informative to reconstruct the original weights, it only consumes 2 bits per element. The number of rows of the co-efficient matrix is also smaller than that of the original weight matrix. These compact representations result in efficient memory compression.

**Binary activation encoding for fast feed-forward propagation**: It has been reported that an inner product between a ternary and binary vector can be computed extremely fast by using three logical operations: AND, XOR, and bit count (Ambai & Sato, 2014). To use this technique, we approximate the activation vector by a weighted sum of binary vectors. This binary encoding must be processed as fast as possible at test-time. To overcome this issue, we use a fast binary encoding method based on a small lookup table.

## 1.1 RELATED WORK

There have been extensive studies on accelerating and compressing deep neural networks, e.g., on an FFT-based method (Mathieu et al., 2014), re-parameterization of a weight matrix (Yang et al., 2015), pruning network connection (Han et al., 2015; 2016), and hardware-specific optimization (Vanhoucke et al., 2011). In the following paragraphs, we only review previous studies that are intimately connected to ours.

It was pointed out by Denil et al. (2013) that network weights have a significant redundancy. Motivated by this fact, researchers have been involved in a series of studies on *matrix/tensor factorization* (Jaderberg et al., 2014; Zhang et al., 2015). In these studies, a weight matrix (or tensor) was factorized by minimizing an approximation error of original weights or activations. Jaderberg et al. (2014) exploited 1-D separable filter decomposition to accelerate feed-forward propagation. Zhang et al. (2015) proposed low-rank approximation based on generalized SVD to compress an entire deep network. Taking into account the lessons learned from these best practices, we also exploit the redundancy of the weights.

There is an another series of studies, *integer decomposition* (Hare et al., 2012; Yuji et al., 2014; Ambai & Sato, 2014), which involved accelerating test-time speed of a classifier by using fast logical operations. Although their contributions are limited to a shallow architecture such as a linear SVM, they achieved a noticeable acceleration. In these approaches, a real-valued weight vector is approximated by a weighted sum of a few binary or ternary basis vectors. To use fast logical operations, they extracted binary features from an image. Hare et al. (2012) and Yuji et al. (2014) exploited binary basis vectors, and Ambai & Sato (2014) investigated a case of ternary basis to improve approximation quality.

In a manner of speaking, our method is a unified framework of *matrix/tensor factorization* and *integer decomposition* reviewed in the above and inherits both their advantages. While the weight matrix is factorized to exploit low-rank characteristics, the basis matrix is restricted to take only three integer values, $\{-1, 0, +1\}$. In contrast to recent binary weighted networks such as XNOR-Net (Rastegari et al., 2016) which quantizes both activations and weights during backpropagation, it is not necessary for our method to change training algorithms at all. We can benefit from recent sophisticated training techniques, e.g. batch normalization (Ioffe & Szegedy, 2015), in combination with our method. Furthermore, our method does not need (iterative) end-to-end retraining which is needed for several previous studies such as network pruning (Han et al., 2015; 2016) and distillation (Hinton et al., 2014).

## 2 NETWORK COMPRESSION MODEL

In this section, we introduce our compression model and discuss time and space complexity. We consider a convolutional layer with a filter size of $w_x \times w_y \times c$, where $w_x$ and $w_y$ are the spacial size and $c$ is a number of input channels. If $w_x = w_y = 1$, we can regard this layer as a fully connected layer. This three dimensional volume is reshaped to form a $D_I$ dimensional vector where $D_I = w_x \times w_y \times c$. The filter weights and biases can be formulated by $\mathbf{W} \in \mathbb{R}^{D_I \times D_O}$ and $\mathbf{b} \in \mathbb{R}^{D_O}$, where $D_O$ is a number of output channels. Let $\mathbf{x} \in \mathbb{R}^{D_I}$ denote an activation vector obtained by

Table 1: Number of operations

| operation | floating point | logical | | |
|---|---|---|---|---|
| | multiply-adds | AND | XOR | bit count |
| original ($\mathbf{W}^\top\mathbf{x}$) | $D_I D_O$ | 0 | 0 | 0 |
| proposed ($\mathbf{C}_w^\top\mathbf{M}_w^\top\mathbf{M}_x\mathbf{c}_x$) | $k_x k_w + k_w D_O$ | $(D_I k_x k_w)/B$ | $(D_I k_x k_w)/B$ | $(D_I k_x k_w)/B$ |

Table 2: Memory consumption. Real value is represented in single precision (32 bits/element).

| variables | original | proposed | | |
|---|---|---|---|---|
| | $\mathbf{W}$ | $\mathbf{M}_w$ | $\mathbf{C}_w$ | $\mathbf{c}_x, b_x$ |
| size (bits) | $32 \cdot D_I D_O$ | $2 \cdot D_I k_w$ | $32 \cdot k_w D_O$ | $32 \cdot (k_x + 1)$ |

vectorizing the corresponding three dimensional volume. In test-time, we need to compute $\mathbf{W}^\top\mathbf{x} + \mathbf{b}$ followed by a non-linear activation function.

In our compressed network, $\mathbf{W}$ is decomposed into two matrices before test-time as follows:

$$\mathbf{W} \approx \mathbf{M}_w \mathbf{C}_w, \tag{1}$$

where $\mathbf{M}_w \in \{-1, 0, +1\}^{D_I \times k_w}$ is a ternary basis matrix, $\mathbf{C}_w \in \mathbb{R}^{k_w \times D_O}$ is a co-efficient matrix, and $k_w$ is the number of basis vectors, respectively. Since $\mathbf{M}_w$ only takes the three values, it consumes only 2 bits per element. Setting a sufficiently small value to $k_w$ further reduces total memory consumption. From the viewpoint of approximation quality, it should be noted that a large number of elements in $\mathbf{W}$ takes close to zero values. To fit them well enough, a zero value must be included in the basis. The ternary basis satisfies this characteristic. In practice, the ternary basis gives better approximation than the binary basis, as we discuss in Section 3.

The activation vector $\mathbf{x}$ is also factored to the following form:

$$\mathbf{x} \approx \mathbf{M}_x \mathbf{c}_x + b_x \mathbf{1}, \tag{2}$$

where $\mathbf{M}_x \in \{-1, +1\}^{D_I \times k_x}$ is a binary basis matrix, $\mathbf{c}_x \in \mathbb{R}^{k_x}$ is a real-valued co-efficient vector, $b_x \in \mathbb{R}$ is a bias, and $k_x$ is the number of basis vectors, respectively. Since elements of $\mathbf{x}$ are often biased, e.g., activations from ReLU take non-negative values and have a non-zero mean, $b_x$ is added to this decomposition model. While $\mathbf{c}_x$ and $b_x$ reflect a range of activation values, $\mathbf{M}_x$ determines approximated activation values within the defined range. This factorization must be computed at test-time because the intermediate activations depend on an input to the first layer. However, in practice, factorizing $\mathbf{x}$ into $\mathbf{M}_x, \mathbf{c}_x$, and $b_x$ requires an iterative optimization, which is very slow. Since a scale of activation values within a layer is almost similar regardless of $\mathbf{x}$, we pre-computed canonical $\mathbf{c}_x$ and $b_x$ in advance and only optimized $\mathbf{M}_x$ at test-time. As we discuss in Section 4, an optimal $\mathbf{M}_x$ under fixed $\mathbf{c}_x$ and $b_x$ can be selected using a lookup table resulting in fast factorization.

Substituting Eqs.(1) and (2) into $\mathbf{W}^\top\mathbf{x} + \mathbf{b}$, approximated response values are obtained as follows:

$$\mathbf{W}^\top\mathbf{x} + \mathbf{b} \approx (\mathbf{M}_w \mathbf{C}_w)^\top (\mathbf{M}_x \mathbf{c}_x + b_x \mathbf{1}) + \mathbf{b} = \mathbf{C}_w^\top \mathbf{M}_w^\top \mathbf{M}_x \mathbf{c}_x + b_x \mathbf{C}_w^\top \mathbf{M}_w^\top \mathbf{1} + \mathbf{b}. \tag{3}$$

A new bias $b_x \mathbf{C}_w^\top \mathbf{M}_w^\top \mathbf{1} + \mathbf{b}$ in Eq.(3) is pre-computable in advance, because $\mathbf{C}_w, \mathbf{M}_w$, and $b_x$ are fixed at test-time. It should be noted that $\mathbf{M}_w^\top \mathbf{M}_x$ is a multiplication of the ternary and binary matrix, which is efficiently computable using three logical operations: XOR, AND, and bit count, as previously investigated (Ambai & Sato, 2014). After computing $\mathbf{M}_w^\top \mathbf{M}_x$, the two co-efficient components, $\mathbf{c}_x$ and $\mathbf{C}_w$, are multiplied from the right and left in this order. Since $\mathbf{c}_x$ and $\mathbf{C}_w$ are much smaller than $\mathbf{W}$, the total number of floating point computations is drastically reduced.

The time and space complexity are summarized in Tables 1 and 2. As can be seen from Table 1, most of the floating operations are replaced with logical operations. In this table, $B$ means the bit width of a variable used in the logical operations, e.g., $B = 64$ if a type of *unsigned long long* is used in C/C++ language. Table 2 suggests that if $k_w$ is sufficiently smaller than $D_I$ and $D_O$, the total size of $\mathbf{M}_w$ and $\mathbf{C}_w$ is reduced compared to the original parameterization.

---

**Algorithm 1** Decompose $\mathbf{W}$ into $\mathbf{M}_w$ and $\mathbf{C}_w$

---

**Require:** $\mathbf{W}$, $k_w$
**Ensure:** factorized components $\mathbf{M}_w$ and $\mathbf{C}_w$.
 1: $\mathbf{R} \leftarrow \mathbf{W}$
 2: **for** $i \leftarrow 1$ to $k_w$ **do**
 3:     Initialize $\mathbf{m}_w^{(i)}$ by three random values $\{-1, 0, +1\}$.
 4:     Minimize $||\mathbf{R} - \mathbf{m}_w^{(i)} \mathbf{c}_w^{(i)}||_F^2$ by repeating the following two steps until convergence.
 5:         [Step 1] $\mathbf{c}_w^{(i)} \leftarrow \mathbf{m}_w^{(i)\top} \mathbf{R} / \mathbf{m}_w^{(i)\top} \mathbf{m}_w^{(i)}$
 6:         [Step 2] $m_{ij} \leftarrow \underset{\alpha \in \{-1, 0, +1\}}{\arg\min} ||\mathbf{r}_j - \alpha \mathbf{c}_w^{(i)}||_2^2$, for $j = 1, \cdots, D_I$
 7:     $\mathbf{R} \leftarrow \mathbf{R} - \mathbf{m}_w^{(i)} \mathbf{c}_w^{(i)}$
 8: **end for**

---

## 3 Ternary weight decomposition

To factorize $\mathbf{W}$, we need to solve the following optimization problem.

$$J_w = \min_{\mathbf{M}_w, \mathbf{C}_w} ||\mathbf{W} - \mathbf{M}_w \mathbf{C}_w||_F^2. \tag{4}$$

However, the ternary constraint makes this optimization very difficult. Therefore, we take an iterative approach that repeats rank-one approximation one by one, as shown in Algorithm 1. Let $\mathbf{m}_w^{(i)} \in \{-1, 0, +1\}^{D_I \times 1}$ denote an $i$-th column vector of $\mathbf{M}_w$ and $\mathbf{c}_w^{(i)} \in \mathbb{R}^{1 \times D_O}$ denote an $i$-th row vector of $\mathbf{C}_w$. Instead of directly minimizing Eq. (4), we iteratively solve the following rank-one approximation,

$$J_w^{(i)} = \min_{\mathbf{m}_w^{(i)}, \mathbf{c}_w^{(i)}} ||\mathbf{R} - \mathbf{m}_w^{(i)} \mathbf{c}_w^{(i)}||_F^2, \tag{5}$$

where $\mathbf{R}$ is a residual matrix initialized by $\mathbf{W}$. We applied alternating optimization to obtain $\mathbf{m}_w^{(i)}$ and $\mathbf{c}_w^{(i)}$. If $\mathbf{m}_w^{(i)}$ is fixed, $\mathbf{c}_w^{(i)}$ can be updated using a least squares method, as shown in line 5 of Algorithm 1. If $\mathbf{c}_w^{(i)}$ is fixed, $m_{ij}$, the $j$-th element of $\mathbf{m}_w^{(i)}$, can be independently updated by exhaustively verifying three choices $\{-1, 0, +1\}$ for each $j = 1, \cdots, D_I$, as shown in line 6 of Algorithm 1, where $\mathbf{r}_j$ is a $j$-th row vector of $\mathbf{R}$. After the alternating optimization is converged, $\mathbf{R}$ is updated by subtracting $\mathbf{m}_w^{(i)} \mathbf{c}_w^{(i)}$ and passed to the next $(i+1)$-th iteration. Comparison of binary constraints with ternary constraints can be seen in Appendix A.

## 4 Binary activation encoding

Binary decomposition for a given activation vector $\mathbf{x}$ can be performed by minimizing

$$J_x(\mathbf{M}_x, \mathbf{c}_x, b_x; \mathbf{x}) = ||\mathbf{x} - (\mathbf{M}_x \mathbf{c}_x + b_x \mathbf{1})||_2^2. \tag{6}$$

In contrast to the case of decomposing $\mathbf{W}$, a number of basis vectors $k_x$ can be set to a very small value (from 2 to 4 in practice) because $\mathbf{x}$ is not a matrix but a vector. This characteristic enables an exhaustive search for updating $\mathbf{M}_x$. Algorithm 2 is an alternating optimization with respect to $\mathbf{M}_x$, $\mathbf{c}_x$, and $b_x$. By fixing $\mathbf{M}_x$, we can apply a least squares method to update $\mathbf{c}_x$ and $b_x$ (in lines 3-4 of Algorithm 2). If $\mathbf{c}_x$ and $b_x$ are fixed, $\mathbf{m}_x^{(j)}$, the $j$-th row vector of $\mathbf{M}_x$, is independent of any other $\mathbf{m}_x^{(j')}$, $j' \neq j$. We separately solve $D_I$ sub-problems formulated as follows:

$$\mathbf{m}_x^{(j)} = \underset{\boldsymbol{\beta} \in \{-1, +1\}^{1 \times k_x}}{\arg\min} (x_j - (\boldsymbol{\beta} \mathbf{c}_x + b_x))^2, \quad j = 1, \cdots, D_I, \tag{7}$$

where $x_j$ is a $j$-th element of $\mathbf{x}$. Since $k_x$ is sufficiently small, $2^{k_x}$ possible solutions can be exhaustively verified (in line 5 of Algorithm 2).

Our method makes this decomposition faster by pre-computing canonical $\mathbf{c}_x$ and $b_x$ from training data and only optimizing $\mathbf{M}_x$ at test-time using lookup table. This compromise is reasonable because of the following two reasons: (1) scale of activation values is similar regardless of vector elements

---

**Algorithm 2** Decompose $\mathbf{x}$ into $\mathbf{M}_x$, $\mathbf{c}_x$, and $b_x$

---

**Require:** $\mathbf{x}$, $k_x$
**Ensure:** factorized components $\mathbf{M}_x$, $\mathbf{c}_x$, and $b_x$.
 1: Initialize $\mathbf{M}_x$ by three random values $\{-1, +1\}$.
 2: Minimize $||\mathbf{x} - (\mathbf{M}_x \mathbf{c}_x + b_x \mathbf{1})||_2^2$ by repeating the following two steps until convergence.
 3: [Step 1] Update $\mathbf{c}_x$ and $b_x$ using a least squares method.
 4: $\mathbf{c}_x \leftarrow (\mathbf{M}_x^\top \mathbf{M}_x)^{-1} \mathbf{M}_x^\top (\mathbf{x} - b_x \mathbf{1}), \quad b_x \leftarrow \mathbf{1}^\top (\mathbf{x} - \mathbf{M}_x \mathbf{c}_x)/D_I$
 5: [Step 2] Update $\mathbf{m}_x^{(j)}$ for each $j = 1, \cdots D_I$ by an exhaustive search that minimizes Eq.(7).

---

within a layer, and (2) $\mathbf{c}_x$ and $b_x$ reflect a scale of approximated activation values. Knowing these properties, $\mathbf{c}_x$ and $b_x$ are obtained by minimizing $J_x(\hat{\mathbf{M}}_x, \mathbf{c}_x, b_x; \hat{\mathbf{x}})$ ,where $\hat{\mathbf{x}}$ is constructed as follows. First, $N_\mathcal{T}$ different activation vectors $\mathcal{T} \in \{\mathbf{x}_i\}_{i=1}^{N_\mathcal{T}}$ are collected from randomly chosen $N_\mathcal{T}$ training data. Second, $n$ elements are randomly sampled from $\mathbf{x}_i$. The sampled $nN_\mathcal{T}$ elements are concatenated to form a vector $\hat{\mathbf{x}} \in \mathbb{R}^{nN_\mathcal{T}}$. We use $\mathbf{c}_x$ and $b_x$ as constants at test-time, and discard $\hat{\mathbf{M}}_x$.

At test-time, we only need to solve the optimization of Eq. (7) for each $x_j$. This can be regarded as the nearest neighbour search in one-dimensional space. We call $\boldsymbol{\beta}\mathbf{c}_x + b_x$ a prototype. There are $2^{k_x}$ possible prototypes because $\boldsymbol{\beta}$ takes $2^{k_x}$ possible combinations. The nearest prototype to $x_j$ and an optimal solution $\mathbf{m}_x^{(j)}$ can be efficiently found using a lookup table as follows.

**Preparing lookup table**: We define $L$ bins that evenly divide one-dimensional space in a range from the smallest to largest prototype. Let $\hat{x}_l$ denote a representative value of the $l$-th bin. This is located at the center of the bin. For each $\hat{x}_l$, we solve Eq. (7) and assign the solution to the bin.

**Activation encoding**: At test-time, $x_j$ is quantized into $L$-levels. In other words, $x_j$ is transformed to an index of the lookup table. Let $p_{\max}$ and $p_{\min}$ denote the largest and smallest prototype, respectively. We transform $x_j$ as follows:

$$q = (L-1)(x_j - p_{\min})/(p_{\max} - p_{\min}) + 1, \tag{8}$$

$$\hat{l} = \min(\max(\lfloor q + 1/2 \rfloor, 1), L). \tag{9}$$

The range from $p_{\min}$ to $p_{\max}$ is linearly mapped to the range from 1 to $L$ by Eq. (8). The term $q$ is rounded and truncated from 1 to $L$ by the max and min function in Eq. (9). If $L$ is sufficiently large, the solution assigned to the $\hat{l}$-th bin can be regarded as a nearly optimal solution because the difference between $x_j$ and the center of the bin $\hat{x}_{\hat{l}}$ becomes very small. We found that $L = 4096$ is sufficient. The time complexity of this encoding is $\mathcal{O}(D_I)$.

## 5 EXPERIMENTS

We tested our method on three different convolutional neural networks: CNN for handwritten digits (LeCun et al., 1998), VGG-16 for ImageNet classification (Simonyan & Zisserman, 2015), and VGG-Face for large-scale face recognition (Parkhi et al., 2015). To compute memory compression rate, a size of $\mathbf{W}$ and a total size of $\mathbf{M}_w$ and $\mathbf{C}_w$ were compared. To obtain a fair evaluation of computation time, a test-time code of forward propagation was implemented without using any parallelization scheme, e.g., multi-threading or SIMD, and was used for both compressed and uncompressed networks. The computation time includes both binary activation encoding and calculation of Eq. (3). We used an Intel Core i7-5500U 2.40-GHz processor.

### 5.1 CNN FOR HANDWRITTEN DIGITS

MNIST is a database of handwritten digits which consists of 60000 training and 10000 test sets of $28 \times 28$ gray-scale images with ground-truth labels from 0 to 9. We trained our CNN by using an example code in MatConvNet 1.0-beta18 (Vedaldi & Lenc, 2015). Our architecture is similar to LeNet-5 (LeCun et al., 1998) but has a different number of input and output channels. Each layer's configuration is shown below:

$$(\text{conv5-20})(\text{maxpool})(\text{conv5-64})(\text{maxpool})(\text{fc1024-640})(\text{relu})(\text{fc640-10})(\text{softmax}), \tag{10}$$

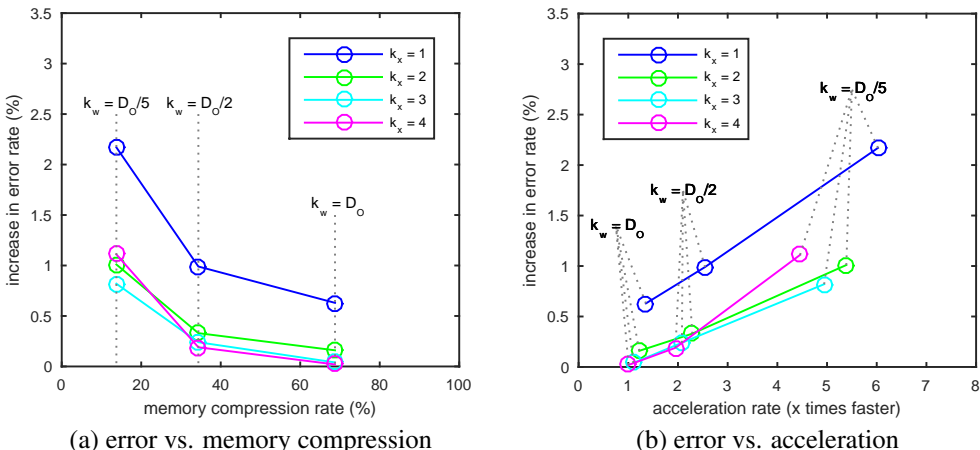

(a) error vs. memory compression (b) error vs. acceleration

Figure 2: Results of MNIST. The first fully connected layer was decomposed.

where the parameters of a convolutional layer are denoted as (conv*<receptive field size>-<number of output channels>*), and parameters of a fully connected layer are denoted as (fc*<number of input channels>-<number of output channels>*). The (maxpool) is $2 \times 2$ subsampling without overlapping. The error rate of this network is $0.86\%$.

We applied our method to the first fully connected layer (fc1024-640) and set $n = 10$ and $N_{\mathcal{T}} = 1000$ to learn $c_x$ and $b_x$ from randomly chosen $nN_{\mathcal{T}}$ activations. The cases of $k_x = 1, 2, 3, 4$ and $k_w = D_O, D_O/2, D_O/5$ were tested. This means that $k_w$ was set to 640, 320, and 128.

Figures 2(a) and (b) show the relationships among the increases in error rates, memory compression rates, and acceleration rates. It was observed that error rates basically improved along with increasing $k_x$ and saturated at $k_x = 4$. It is interesting that $k_x = 2$, only 2 bits per element for encoding an activation $\mathbf{x}$, still achieved good performance. While the smaller $k_w$ achieved better compression and acceleration rate, error rates rapidly increased when $k_w = D_O/5$. One of the well balanced parameters was $(k_x, k_w) = (4, D_O/2)$ which resulted in $1.95\times$ faster processing and a $34.4\%$ memory compression rate in exchange of a $0.19\%$ increase in the error rate.

## 5.2 VGG-16 FOR IMAGENET CLASSIFICATION TASK

A dataset of ILSVRC2012 (Russakovsky et al., 2015) consists of 1.2 million training, 50,000 validation, and 100,000 test sets. Each image represents one of 1000 object categories. In this experiment, we used a network model of VGG-16 (model D in (Simonyan & Zisserman, 2015)) that consists of 13 convolutional layers and 3 fully connected layers followed by a softmax layer. The architecture is shown below:

$$(\text{input}) \cdots (\text{fc25088-4096})(\text{relu})(\text{fc4096-4096})(\text{relu})(\text{fc4096-1000})(\text{softmax}), \qquad (11)$$

where layers before the first fully connected layer are omitted.

First, all three fully connected layers were compressed with our algorithm. We set $n = 10$ and $N_{\mathcal{T}} = 1000$ to learn $c_x$ and $b_x$ from randomly chosen $nN_{\mathcal{T}}$ activations. The cases of $k_x = 2, 3, 4$ and $k_w = D_O/2, D_O/4, D_O/8, D_O/16$ were tested. The case of $k_x = 1$ was omitted because this setting resulted in a very high error rate. Note that each of the fully connected layers has different $D_O$. The $k_w$ was independently set for each layer according to its $D_O$. The top-5 error rates were evaluated on the validation dataset. The top-5 error rate of the original network is $13.4\%$.

The three lines with circles in Figure 3 show these results. It should be noted that much higher acceleration rates and smaller compression rates with small loss of accuracies were achieved than the case of the network for MNIST. Interestingly, the case of $k_w = D_O/4$ still performed well due to the low-rank characteristics of weights in the VGG-16 network.

Although the error rates rapidly increased when $k_w$ took much smaller values, we found that this could be improved by tuning $k_w$ of the third layer. More specifically, we additionally tested the

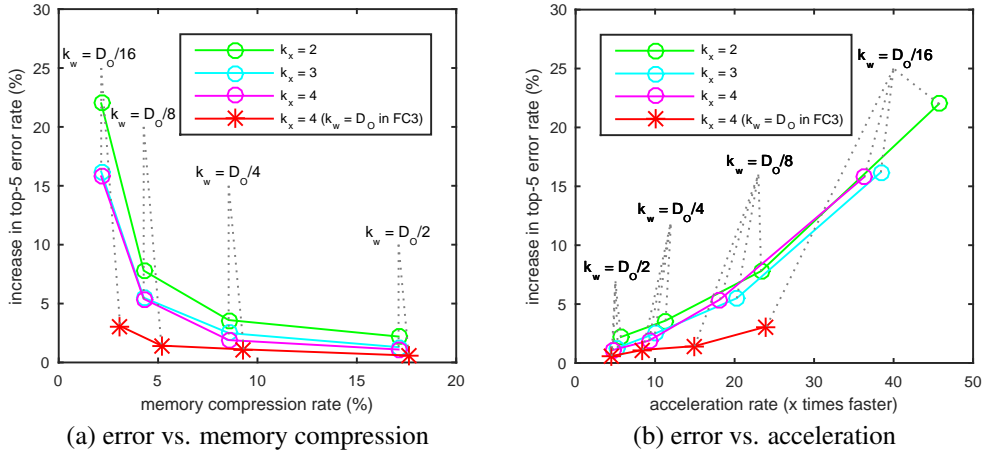

(a) error vs. memory compression    (b) error vs. acceleration

Figure 3: Results of VGG-16. The last three fully connected layers were decomposed.

Table 3: Best balanced parameters for decomposing three fully connected layers of VGG-16.

| Top-5 error (%) | Original 13.4 | | | | Proposed 14.8 | | | |
|---|---|---|---|---|---|---|---|---|
| | MBytes | msec | $k_w$ | $k_x$ | MBytes | ratio | msec | ratio |
| fc25088-4096 | 392.0 | 142.4 | $D_O/8$ | 4 | 11.1 | 2.8% | 6.1 | 23.5× |
| fc4096-4096 | 64.0 | 22.8 | $D_O/8$ | 4 | 8.5 | 13.3% | 3.0 | 7.5× |
| fc4096-1000 | 15.6 | 5.7 | $D_O$ | 4 | 4.8 | 30.7% | 2.3 | 2.5× |
| total | 471.6 | 170.9 | | | 24.4 | **5.2**% | 11.4 | **15.0**× |

Table 4: Reults of decomposing convolutional layers of VGG-16.

| Compressed convolutional layers | 2nd | 2nd-4th | 2nd-6th | 2nd-8th | 2nd-10th |
|---|---|---|---|---|---|
| Increase in top-5 error (%) | 0.37 | 1.64 | 2.79 | 4.13 | 6.42 |
| Acceleration rate of entire network | 1.08× | 1.22× | 1.41× | 1.68× | 2.15× |

following cases. While $k_w$ was set to $D_O/2, D_O/4, D_O/8$, and $D_O/16$ for the first and second layers, $k_w$ was fixed to $D_O$ for the third layer. The $k_x$ was set to 4. This is plotted with a red line in Figure 3. In this way, the memory compression rate and acceleration rate noticeably improved. Setting appropriate parameters for each layer is important to improve the total performance. Table 3 shows the details of the best balanced case in which 15× faster processing and 5.2% compression rate were achieved in exchange of a 1.43% increase in error rate.

Next, we also tested to compress convolutional layers. In this experiment, $k_w$ and $k_x$ were set to $D_O$ and 4. This setting accelerates each of the layers averagely 2.5 times faster. Table 4 shows positions of compressed layers, top-5 errors, and acceleration rates of the entire network. Although $k_w$ and $k_x$ must be larger than those of fully connected layers to avoid error propagation, it is still beneficial for entire acceleration. In summary, while compressing fully connected layers is beneficial for reducing memory, compressing convolutional layers is beneficial for reducing entire computation time.

## 5.3 VGG-FACE FOR FACE RECOGNITION TASK

The VGG-Face (Parkhi et al., 2015) is a model for extracting a face descriptor. It consists of a similar structure to VGG-16. The difference is that VGG-Face has only two fully connected layers, as shown below.

$$(\text{input}) \cdots (\text{fc25088-4096})(\text{relu})(\text{fc4096-4096}). \qquad (12)$$

This network outputs a 4096-dimensional descriptor. We can verify whether two face images are identical, by evaluating the Euclidean distance of two $l^2$-normalized descriptors extracted from

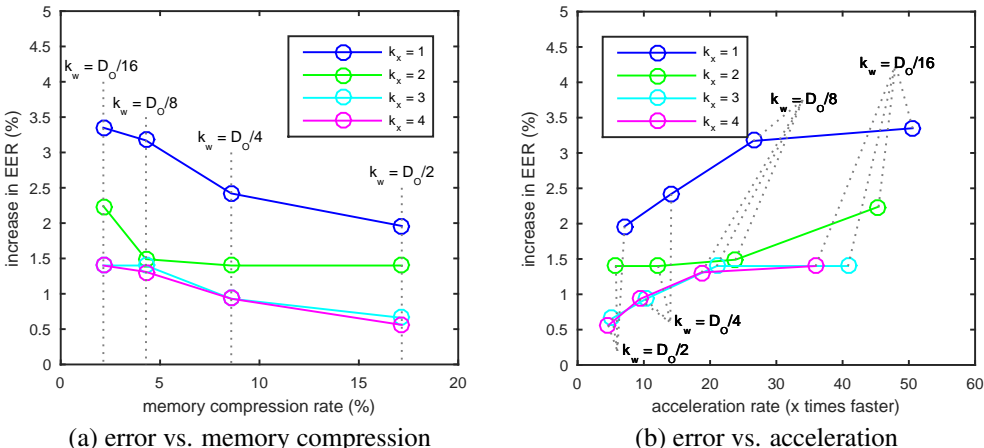

(a) error vs. memory compression (b) error vs. acceleration

Figure 4: Results of VGG-Face. The last two fully connected layers were decomposed.

Table 5: Reults of decomposing convolutional layers of VGG-Face.

| Compressed convolutional layers | 2nd | 2nd-4th | 2nd-6th | 2nd-8th | 2nd-10th |
|---|---|---|---|---|---|
| Increase in EER (%) | 0.28 | 0.47 | 0.66 | 0.47 | 0.75 |

them. In our experiment, we did not apply a descriptor embedding technique based on triplet loss minimization (Parkhi et al., 2015). Following the evaluation protocol introduced in a previous paper (Parkhi et al., 2015), we used Labeled Faces in the Wild dataset (LFW) (Huang et al., 2007), which includes 13,233 face images with 5,749 identities. The LFW defines 1200 positive and 1200 negative pairs for testing. We used the 2400 test pairs to compute ROC curve and equal error rate (EER). The EER is defined as an error rate at the ROC operating point where the false positive and false negative rates are equal. The EER of the original network is $3.8\%$.

First, the two fully connected layers were compressed using our algorithm. We set $n = 10$ and $N_{\mathcal{T}} = 1000$ to learn $c_x$ and $b_x$ from randomly chosen $nN_{\mathcal{T}}$ activations. We tested the cases of $k_x = 1, 2, 3, 4$, and $k_w = D_O/2, D_O/4, D_O/8, D_O/16$. Figure 4 reveals an interesting fact that even the fastest and smallest network configuration, $k_x = 1$ and $k_w = D_O/16$, had less impact on the EER, in contrast to the previous ImageNet classification task in which the recognition results were corrupted when $k_x = 1$. This indicates that the 4096-dimensional feature space is well preserved regardless of such coarse discretization of both weights and activations.

Next, we also tested to compress convolutional layers. In this experiment, $k_w$ and $k_x$ were set to $D_O$ and 4 which are the the same setting used in Table 4. Table 5 shows positions of compressed layers and EERs. The acceleration rates were almost the same as the results shown in Table 4. This is because architecture of VGG-face is the same as VGG-16 and we used the same parameter for $k_w$ and $k_x$. Interestingly, compressing multiple layers from 2nd to 10th still preserves the original EER. As can be seen from this table, our method works very well depending on a certain kind of machine learning task.

## 6 CONCLUSION

We proposed a network compression model that consists of two components: ternary matrix decomposition and binary activation encoding. Our experiments revealed that the proposed compression model is available not only for multi-class recognition but also for feature embedding. Since our approach is post-processing for a pre-trained model, it is promising that recent networks designed for semantic segmentation, describing images, stereo matching, depth estimation, and much more can also be compressed with our method. For future work, we plan to improve approximation error further by investigating the discrete optimization algorithm.

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

APPENDIX

## A BINARY VS. TERNARY

Figure 5 illustrates the reconstruction errors of a $4096 \times 1000$ weight matrix of the last fully connected layer in VGG-16 model (Simonyan & Zisserman, 2015). We tested both the binary and ternary constraints on $\mathbf{M}_w$ for comparison. The reconstruction error $J_w$ monotonically decreased along with an increase in $k_w$. It was clear that the ternary basis provided better reconstruction than the binary basis.

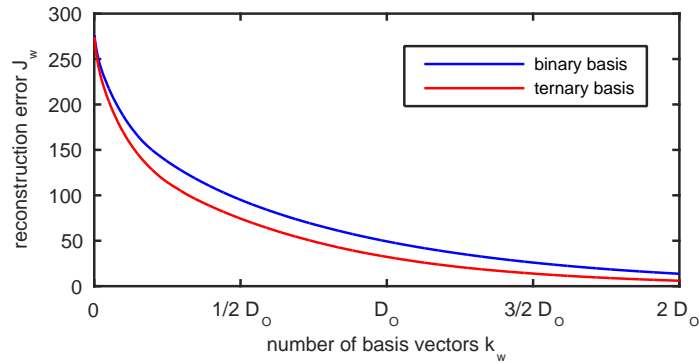

Figure 5: $4096 \times 1000$ weight matrix of last fully connected layer in VGG-16 model (Simonyan & Zisserman, 2015) is decomposed under two different constraints: (blue) $\{-1, +1\}$ and (red) $\{-1, 0, +1\}$.

