# Peer review of "Ternary Weight Decomposition and Binary Activation Encoding for Fast and Compact Neural Network"

_ICLR 2017 — rejected_

[Public Comment · Sungho Shin · 15 Dec 2016]
**Suggestion for missing reference**

I suggest to refer the following two papers.

- Kyuyeon Hwang and Wonyong Sung. "Fixed-point feedforward deep neural network design using weights +1, 0, and −1." 2014 IEEE Workshop on Signal Processing Systems (SiPS). IEEE, 2014.

- Jonghong Kim, Kyuyeon Hwang, and Wonyong Sung. "X1000 real-time phoneme recognition VLSI using feed-forward deep neural networks." 2014 IEEE International Conference on Acoustics, Speech and Signal Processing (ICASSP). IEEE, 2014.

The retrain-based neural network quantization algorithm was first published in these two papers.

Thanks.

[Official Review · AnonReviewer2 · rating 5 · confidence 3 · 16 Dec 2016]
**Clarify my comments**

I do need to see the results in a clear table. Original results and results when compression is applied for all the tasks. In any case, i would like to see the results when the compression is applied to state of the art nets where the float representation is important. For instance a network with 0.5% - 0.8% in MNIST. A Imagenet lower that 5% - 10%. Some of this results are feasible with float representation but probably imposible for restricted representations.

[Official Review · AnonReviewer3 · rating 4 · confidence 4 · 16 Dec 2016]
**Novel quantization method to reduce memory and complexity of pre-trained networks, but benefit over other methods is unclear**

This paper explores a new quantization method for both the weights and the activations that does not need re-training. In VGG-16 the method reaches compression ratios of 20x and experiences a speed-up of 15x. The paper is very well written and clearly exposes the details of the methodology and the results.

My major criticisms are three-fold: for one, the results are not compared to one of the many other pruning methods that are described in section 1.1, and as such the performance of the method is difficult to judge from the paper alone. Second, there have been several other compression schemes involving pruning, re-training and vector-quantization [e.g. 1, 2, 3] that seem to achieve much higher accuracies, compression ratios and speed-ups. Hence, for the practical application of running such networks on low-power, low-memory devices, other methods seem to be much more suited. The advantage of the given method - other then possibly reducing the time it takes to compress the network - is thus unclear. In particular, taking a pre-trained network as a starting point for a quantized model that is subsequently fine-tuned might not take much longer to process then the method given here (but maybe the authors can quantify this?). Finally, much of the speed-up and memory reduction in the VGG-model seems to arise from the three fully-connected layers, in particular the last one. The speed-up in the convolutional layers is comparably small, making me wonder how well the method would work in all-convolutional networks such as the Inception architecture.

[1] Deep Compression: Compressing Deep Neural Networks with Pruning, Trained Quantization and Huffman Coding,

[Official Review · AnonReviewer1 · rating 6 · confidence 3 · 19 Dec 2016]

This paper addresses to reduce test-time computational load of DNNs. Another factorization approach is proposed and shows good results. The comparison to the other methods is not comprehensive, the paper provides good insights.

[Final Decision · Program Chairs · 06 Feb 2017]
**ICLR committee final decision**

The paper presents a method for quantizing neural network weights and activations. The method is not compared to related state-of-the-art quantization techniques, so in the current form the paper is not ready for acceptance.